# Learning Multi-Faceted Prototypical User Interests

**Nhu-Thuat Tran**
Singapore Management University
Singapore
nttran.2020@phdcs.smu.edu.sg

**Hady W. Lauw**
Singapore Management University
Singapore
hadywlauw@smu.edu.sg

## Abstract

We seek to uncover the latent interest units from behavioral data to better learn user preferences under the VAE framework. Existing practices tend to ignore the multiple facets of item characteristics, which may not capture it at appropriate granularity. Moreover, current studies equate the granularity of item space to that of user interests, which we postulate is not ideal as user interests would likely map to a small subset of item space. In addition, the compositionality of user interests has received inadequate attention, preventing the modeling of interactions between explanatory factors driving a user's decision. To resolve this, we propose to align user interests with multi-faceted item characteristics. First, we involve prototype-based representation learning to discover item characteristics along multiple facets. Second, we compose user interests from uncovered item characteristics via binding mechanism, separating the granularity of user preferences from that of item space. Third, we design a dedicated bi-directional binding block, aiding the derivation of compositional user interests. On real-world datasets, the experimental results demonstrate the strong performance of our proposed method compared to a series of baselines.

## 1 Introduction

Preference learning lies at the heart of Collaborative Filtering (CF). A common thread among many CF models Sedhain et al. (2015); Liang et al. (2018); Wang et al. (2019a); He et al. (2020) is rendering a single vector for preference representation, ignoring the fact that user preferences are complex and diverse. Thus, discovering hidden factors behind user preferences could provide insights into what governs consumption patterns of users and thereby boosting the recommendation performance.

Nonetheless, preference learning is considerably challenging because of its unstructured nature. In natural language, one can consider a word or a token as a modular unit. However, in CF, we first need to obtain such 'units', i.e., by deriving multiple vectors, each is a unit, of user's preferences. Yet it is quite elusive what structure and level of granularity these preference units should appropriately be, and more importantly, how to obtain them in unsupervised setting based only on observed behavior data.

One means to derive a preference 'unit' is by grouping related items into a cluster that represents a meaningful interest, then aggregating item representations to produce interest representation and finally, jointly learning item grouping and recommendation under Variational AutoEncoder (VAE) framework Ma et al. (2019); Tran & Lauw (2022); Wang et al. (2023a). Marrying item grouping-based interest derivation with VAE inherits the best of both worlds. For one, multiple item groups increase representation capacity of VAE to capture multiple intentions of users behind consumption behaviors Ma et al. (2019). For another, interest derivation process inherits VAE's strengths, including non-linear probabilistic modeling, multinomial likelihood as a proxy of ranking loss, and information-theoretic regularization term, which are shown to boost the recommendation accuracy Liang et al. (2018).

Despite showing improvement to a degree, existing VAE-based recommendation models involving item grouping for interest derivation has several shortcomings. First, these studies ignore that items may be grouped by arbitrary characteristics, owing to their many facets. For example, as depicted in Figure 1, shoes can be grouped by multiple facets, e.g., *brand*, *color* or *top height*. Thus, a more fine-grained structure that focuses on each facet would better reflect item space structure. Second, the assumption that the number of interests per user equals the number of item groups is sub-optimal as a

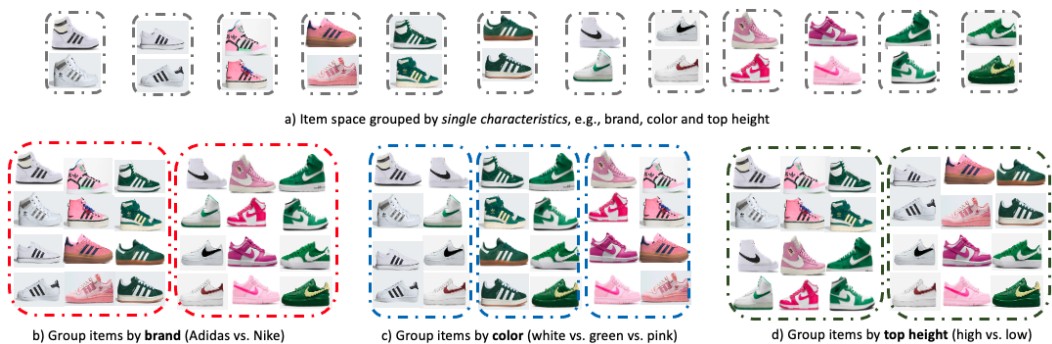

Figure 1: Discovering item space structure under multiple facets. In Figure (a), shoe space is a combination of facets, which requires as many as 12 clusters. Figures (b), (c), (d), each divides shoe space by a single facet. The total number of clusters is 7, which is slightly more than half of that in Figure (a).

user only interacts with small subset of items among the whole item space. Third, the compositionality of user interests has received less attention, which may not capture the complexity of user's interest.

Towards addressing these shortcomings, we introduce FACETVAE, which stands for FACETed Variational AutoEncoder, distinguishing itself by three key innovations. *Firstly*, to fully reveal the structure of item space so as to better align with user interests, we discover multiple groups underlying item space along multiple dimensions via prototype-based representation. For example, three dimensions to group items in Figure 1 are *brand*, *color* or *top height*. *Secondly*, under each dimension, we aggregate representations of user-adopted items belonging to a specific group, e.g., shoes assigned to Nike group under *band* dimension. The output is an array of *low-level* user interests towards multiple item characteristics. These *low-level* interests are ingredients to construct compositional (*high-level*) user interests. By this design, we separate the number of user interests from that of item space granularity. *Thirdly*, we introduce a novel bi-directional binding block to better derive compositional user interests, which includes prototype competition to explain low-level interests and low-level interests competition to attend high-level interests.

**Contributions.** The primary contributions of this paper are three-fold. *First*, we propose to discover item space structure under a multiple-facet lens to better derive compositional user interests from uncovered item characteristics. *Second*, we introduce FACETVAE to improve VAE-based preference learning. Our novelties include identifying multi-faceted item structure, binding compositional user interests and bi-directional user interest binding. *Third*, we extensively conduct experiments on real-world datasets to demonstrate the state-of-the-art recommendation accuracy of FACETVAE. In addition, we provide a qualitative analysis to ease the understanding of FACETVAE's inner working.

## 2 RELATED WORK

**Multi-interest user modeling.** The most popular method is item grouping, which has been explored for Collaborative Filtering (CF) Ma et al. (2019), CF with side information Tran & Lauw (2022); Wang et al. (2023a) and sequential recommendation Li et al. (2019); Cen et al. (2020); Xiao et al. (2020); Tan et al. (2021a); Zhang et al. (2022); Wang & Shen (2022); Wang et al. (2022a). The second method relies on representation learning on graph, i.e., DGCF Wang et al. (2020) divides user (item) representation into $K$ factors then uses routing mechanism to aggregate information from neighbors to obtain multiple interests of user (item), DCCF Ren et al. (2023) leverages intent prototypes to aggregate global context information at a graph embedding layer. The third method projects user embedding vector into multiple spaces, each captures one aspect of their preferences Weston et al. (2013); Tan et al. (2021b); Bao et al. (2022). Our work falls into item grouping-based approach yet is more generalized by multi-faceted disentangling and binding compositional user interests.

**Disentangled representation learning.** The idea is to discover the factors of variation underlying data Bengio et al. (2013). This principle has been applied in CF Ma et al. (2019); Wang et al. (2020); Ren et al. (2023), sequential recommendation Ma et al. (2020); Zheng et al. (2021; 2022), side information-aware recommendation Zhang et al. (2020); Tran & Lauw (2022); Wang et al. (2023a;b),

citation recommendation Wang et al. (2022c), bundle recommendation Zhao et al. (2022). Our work is close to MacridVAE Ma et al. (2019), DGCF Wang et al. (2020) and DCCF Ren et al. (2023). The proposed FACETVAE and MacridVAE have in common micro-disentanglement, which does not appear in DGCF and DCCF. FACETVAE generalizes MacridVAE by disentangling item space under multi-faceted lens. Furthermore, FACETVAE binds low-level user's interests into compositional ones, which is another novelty. FACETVAE also relates to disentangling representation's dimensions Higgins et al. (2017); Kim & Mnih (2018); Chen et al. (2018); Locatello et al. (2019). Not only do we disentangle single vector representation, but we also discover multiple clusters of item space under multiple facets.

**Multiple clusters discovery.** Prior arts on discovering multiple clusters under data Jain et al. (2008); Zhao et al. (2017); Wang et al. (2018; 2019b); Yao et al. (2019) optimize a clustering objective, merely aiming at grouping data points and therefore, they are widely different from ours. For one, our clustering process is guided by a recommendation objective. For another, prior works assume each data point is from a concept while we assume that each item interacts with different concepts (prototypes of facets) to derive representation for a data point (a user).

**Binding problem in neural network.** The idea is to group relevant low-level features into semantically meaningful high-level ones Greff et al. (2020), which has some commonality with ours. A related problem is object-centric learning, aiming to derive representations for multiple objects in the input image Locatello et al. (2020); Singh et al. (2022); Chang et al. (2022); Singh et al. (2023); Jia et al. (2023). Our motivation differs from those of these mentioned works as we aim to derive user's interests from their adoptions for collaborative filtering task.

## 3 PRELIMINARIES

Our problem follows the typical settings of collaborative filtering, including $M$ users and $N$ items. Let $y_i^u = 1$ be an observed interaction between user $u$ and item $i$ and $y_i^u = 0$ means no interaction has been recorded between the two. Let $\mathbf{x}^u = \{i : y_i^u = 1\}$ be the set of interacted items of $u$. The target is to predict the probability that a user $u$ will interact with an item $i$ based on $u$'s past interactions.

**Problem Formulation.** There exist complex patterns driving user's item adoption behaviors. Thus, uncovering these hidden explanatory factors would not only enhance interpretability of user preferences but also provides pathway to improve recommendation task. As such, we seek for a set of $K$ vectors $\mathbf{z}^u = \{\mathbf{z}_k^u \in \mathbb{R}^d\}_{k=1}^K$ representing $K$ interests of user and $K$ is a pre-defined number.

**Multi-interest modeling under VAE framework.** We briefly describe a representative work MacridVAE Ma et al. (2019) for illustration. MacridVAE involves three main steps, *item grouping*, *user interest aggregating*, and *decoding and learning*. *First*, a set of $K$ prototypes $\mathbf{m} \in \mathbb{R}^{K \times d}$ is employed to group $N$ items $\mathbf{T} \in \mathbb{R}^{N \times K}$ into $K$ groups, generating assignment matrix $\mathbf{C} \in \mathbb{R}^{N \times K}$. Prototypes are randomly initialized and learned in data-driven manner. *Second*, given $\mathbf{C}$ and context matrix $\mathbf{E} \in \mathbb{R}^{N \times d^{enc}}$, MacridVAE aggregates a user's adopted items belonging to a specific cluster to produce a set of vectors $\{\mathbf{h}_k^u\}_{k=1}^K$ [1]. Then an interest vector is sampled from Gaussian distribution with parameters estimated via a neural network $\varphi$, i.e., $(\mathbf{a}_k^u, \mathbf{b}_k^u) = \varphi(\mathbf{h}_k^u) \; \forall k = 1, 2, ..., K$.

$$\mathbf{C} = \phi(\frac{\mathbf{T} \cdot \mathbf{m}^T}{\tau \cdot ||\mathbf{T}||_2 \cdot ||\mathbf{m}||_2}) \implies (\mathbf{a}_k^u, \mathbf{b}_k^u) = \varphi(\frac{\sum_{i \in \mathbf{x}^u} \mathbf{C}_{ik} \cdot \mathbf{E}_i}{\sqrt{\sum_{i \in \mathbf{x}^u}(\mathbf{C}_{ik})^2}}) \implies \mathbf{z}_k^u \sim \mathcal{N}(\frac{\mathbf{a}_k^u}{||\mathbf{a}_k^u||_2}, [diag(\sigma_0 \cdot \exp(-\frac{1}{2}\mathbf{b}_k^u))]^2)$$

$\phi$ is Gumbel-Softmax Jang et al. (2017); Maddison et al. (2017) to approximate one-hot vector, i.e., if item $i$ belongs to cluster $k$ then $\mathbf{C}_{ik} \approx 1$ and $\mathbf{C}_{ij} \approx 0 \; \forall j \neq k$. $\tau$ is a temperature hyper-parameter to obtain more skewed distribution. $\sigma_0$ is a hyper-parameter with value around 0.1. *Thirdly*, decoder predicts score $r(\mathbf{z}_k^u)$ then normalize to obtain probability of user-item interaction.

$$r(\mathbf{z}_k^u) = exp(\frac{\mathbf{z}_k^u \cdot (\mathbf{T}_i)^T}{\tau_{dec} \cdot ||\mathbf{z}_k^u||_2 \cdot ||\mathbf{T}_i||_2}) \implies p(y_i^u | \mathbf{x}^u, \mathbf{C}) = \frac{\sum_{k=1}^K \mathbf{C}_{ik} r(\mathbf{z}_k^u)}{\sum_{i=1}^N \sum_{k=1}^K \mathbf{C}_{ik} r(\mathbf{z}_k^u)}$$

Assignment matrix $\mathbf{C}$ is used to weight the prediction, i.e., if item $i$ probably belongs to $k^{th}$ cluster, the predicted score of item $i$ by $k^{th}$ interest will be given the corresponding weight. The final prediction score is summed over predictions of $K$ user interests. Finally, MacridVAE's learning objective includes two terms: cross-entropy loss to reconstruct observed user-item interactions and Kullback-Leibler divergence to regularize variational distribution of user interests with prior distribution.

---

[1] We skip bias vector when calculating $\mathbf{h}_k^u$ to ease understanding.

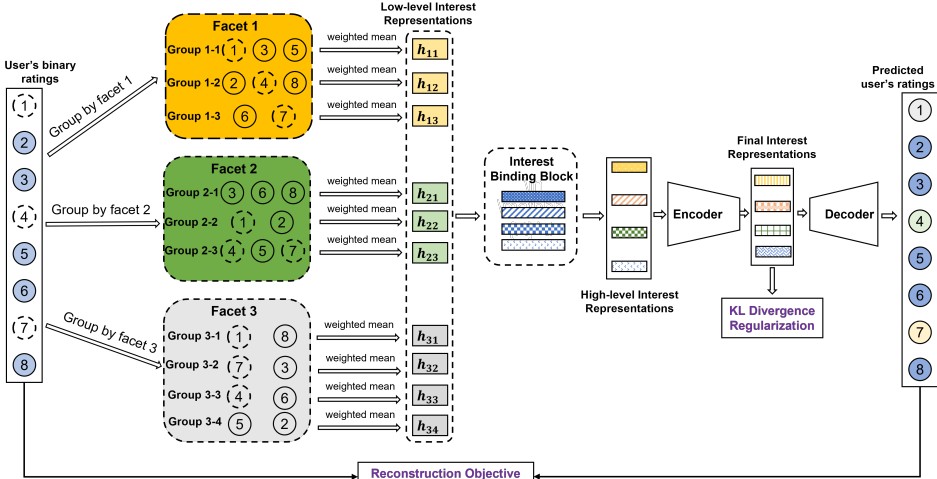

Figure 2: Architecture of FACETVAE. Input includes item IDs with shaded circles are adopted items and dashed ones are not adopted items. FACETVAE groups items into multiple clusters under multiple facets to derive low-level interests (dashed items are not considered), then composes them into high-level one by binding block. KL divergence regularization is to disentangle representation dimensions.

**Limitations.** Existing VAE methods that seek to discover user interests via prototype-based item grouping have three main shortcomings. *First*, multiple facets underlying item space are inadequately uncovered. As in Figure 1, it requires to uncover three facets (*color*, *brand*, *top height*) as well as facet-wise item groups, e.g., Nike vs. Adidas under *brand* facet, to fully model item space. *Second*, the assumption that the granularity of item space equals to that of user interests, causes a dilemma. On the one hand, a large number of item groups is required to model item space structure yet it is exaggerated to model user interests as a user often only consumes a small subset of item space. On the other hand, while a small number of item groups is reasonable for modeling user interests, it is insufficient to capture item space structure, which might be multi-faceted as in Figure 1. *Third*, the complexity of user interests has received inadequate attention. As multiple factors may influence a user's decision, the compositionality of user interests is crucial when deriving interest representations.

## 4 MULTI-FACETED PROTOTYPICAL USER INTERESTS LEARNING

Figure 2 illustrates FACETVAE, a solution for the mentioned shortcomings. The key innovations are *i)* discovering item space structure along multiple facets; *ii)* binding compositional user interests from discovered item characteristics; *iii)* composing user interests via a bi-directional binding block.

### 4.1 MULTI-FACETED ITEM SPACE STRUCTURE DISCOVERING

We aim at capturing the granularity of item space to better infer user preferences, e.g., uncovering three facets (*color*, *brand*, *top height*) and their corresponding granularity, e.g., Adidas vs. Nike shoes under *brand* facet in Figure 1. These facets are assumed to be latent and to be uncovered unsupervisedly.

To realize, we infer a set of $F$ matrices $\mathbf{C} = \{\mathbf{C}_f\}_{f=1}^{F}$, assuming $F$ facets underlying item space. Each $\mathbf{C}_f \in \mathbb{R}^{N \times J}$ represents clustering of $N$ items to $J$ clusters under facet $f$. Here, we assume $J$ item groups under each facet. It is straightforward to apply our method when the number of item groups under each facet differs. Moreover, due to the inaccessibility to the prior information about facets behind data, we have assumed uniform facet distribution. This naturally holds as facets underlying item space simultaneously exists. Multi-faceted item grouping brings two salient benefits.

*Efficiently modeling item space.* As illustrated in Figure 1 (a), one can use single facets to group items. However, it requires the exaggerated number of groups, which scales exponentially with the number of facets and the number of clusters per facet, i.e., $J_1 \times J_2 \times ... \times J_F$ assuming $J_f$ groups under $f^{th}$ facet. Contrarily, multi-faceted item grouping only requires $J_1 + J_2 + ... + J_F$ groups, a linear function of $J_f$.

*Composition of multiple facets.* As being driven by many factors, user interests might include multiple item characteristics, e.g., *white low top Nike shoes* or *green high top Adidas shoes*. Multi-faceted item grouping enables composing such complicated interests from discovered items characteristics.

Concretely, we involve $\mathbf{P} \in \mathbb{R}^{F \times J \times d}$ as the prototype collection of $F$ facets, $\mathbf{P}_f \in \mathbb{R}^{J \times d}$ be the prototypes under $f^{th}$ facet. Prototypes are expected to convey a specific characteristic of item space. For example, under *brand* facet, there are two prototypes Adidas and Nike. We assume these prototypes are latent. They are updated in a data-driven manner and then used to aggregate items having the same characteristic. In real-world scenarios where item category knowledge is available, one could embed semantic information into these prototypes by classifying them into the corresponding category labels. We perform facet-wise item grouping by estimating the assignment score of item $i$ to cluster $j$ under $f$ as

$$\mathbf{C}_{fij} = \phi([\mathbf{s}_{fi1}, \mathbf{s}_{fi2}, ..., \mathbf{s}_{fiJ}]); \quad \mathbf{s}_{fij} = \mathbf{T}_i^T \mathbf{P}_{fj} / (\tau \cdot ||\mathbf{T}_i||_2 \cdot ||\mathbf{P}_{fj}||_2) \tag{1}$$

Following Ma et al. (2019), $\phi$ is Gumbel-Softmax Maddison et al. (2017); Jang et al. (2017) to approximate one-hot vector of the cluster distribution $\mathbf{C}_{fi} \in \mathbb{R}^J$ of item $i$ under facet $f$. $\mathbf{s}_{fij}$ is based on cosine similarity between item embedding vector $\mathbf{T}_i \in \mathbb{R}^d$ and prototype $\mathbf{P}_{fj} \in \mathbb{R}^d$ and $|| \cdot ||_2$ is $L_2$ norm. Using cosine similarity prevents all items are associated to a cluster with highest magnitude $||\mathbf{P}_{fj}||_2$. $\tau$ is the temperature to concentrate the weight to the most similar cluster. By calculating Equation 1 for $F$ facets and $N$ items, we obtain assignment score matrix $\mathbf{C} \in \mathbb{R}^{F \times N \times J}$.

The output of Equation 1 satisfies $\sum_{j=1}^{J} \mathbf{C}_{fij} = 1$ and $\mathbf{C}_{fij} \geq 0$. This naturally creates a competition between $J$ clusters, i.e., they compete to other $J - 1$ clusters for attending to an item $i$. For example, under *color* facet in Figure 1, if $green$ shoes are tied to cluster $J$, i.e., the assignment score of $green$ shoes to cluster $J$ is high, which results in lower assignment scores of $green$ shoes to other clusters. Therefore, the nature of this competition enables grouping related items into meaningful clusters.

## 4.2 BINDING COMPOSITIONAL USER INTEREST REPRESENTATIONS

This section presents the derivation of user interests from the uncovered item space structure.

**Low-level user interest representation.** Intuitively, if a user adopts an item belonging to a specific group, it is likely that they are interested in item characteristic captured by that group, motivating us to derive user interest representation based on item group clues. The output is called *low-level* as it is supposed to capture a single item characteristic, e.g., Nike shoes under brand facet.

Let $\mathbf{h}_{fj}^u \in \mathbb{R}^{d^{enc}}$ be low-level interest of user $u$ towards characteristic $j$ under facet $f$. $\mathbf{h}_{fj}^u$ is derived from user $u$'s interacted items given cluster distribution $\mathbf{C}_f$, bias $\mathbf{b}^{enc} \in \mathbb{R}^{d^{enc}}$, activation function $\gamma_0$

$$\mathbf{h}_{fj}^u = \gamma_0 (\sum_{i \in \mathbf{x}^u} \mathbf{C}_{fij} \cdot \mathbf{E}_i / \sqrt{Z} + \mathbf{b}^{enc}) \quad \text{with} \quad Z = \sum_{i \in \mathbf{x}^u} (\mathbf{C}_{fij})^2 \tag{2}$$

$\mathbf{E}_i \in \mathbb{R}^{d^{enc}}$ is the context vector used to derive interests. For simplicity, a default setting is $d^{enc} = d$. $\mathbf{H}^u = \{\mathbf{h}_{fj}^u\}_{f=1, j=1}^{F, J} \in \mathbb{R}^{F \times J \times d^{enc}}$ captures $J$ low-level user's interests underlying $F$ facets.

**High-level user interest representation.** A user's decision is driven by multiple factors, e.g., *brand* and/or *color* when they buy a new pair of shoes. Thus, modeling the composition of factors having influence on a user's decision is essential. We regard $\mathbf{H}^u$ as the ingredients to compose high-level user's interests. Formally, we employ a set of $K$ prototypes denoted by $\mathbf{Q} \in \mathbb{R}^{K \times d}$. $\mathbf{Q}$ will retrieve low-level user interests from $\mathbf{H}^u$ and then compose them into $K$ high-level user interests.

Nevertheless, this is non-trivial task for a couple of reasons. For one, it demands a dedicated mechanism that wisely binds low-level user interests. For example, *low top* and *high top* shoes should be assigned to two different high-level interests as a pair of shoes do not simultaneously contain these two characteristics. For another, high-level interests are required to be distinct to capture the diversity of user preferences and alleviate the negative effect of noisy and redundant interests.

**Bi-directional binding block.** We present *bi-directional* binding mechanism as a solution. Not only does it bind high-level user interests over low-level ones, but also it enables competition between

low-level interests to attend to high-level counterparts. Given $\mathbf{Q}$ and $\mathbf{H}^u$, our binding block works as

$$\mathbf{A}_{fjk}^u = \frac{1}{2}[\underset{1,2,...,K}{softmax}(sim(\mathbf{h}_{fj}^u, w(\mathbf{Q}_k))/\tau_0) + \underset{1,2,...,J}{softmax}(sim(\mathbf{h}_{fj}^u, w(\mathbf{Q}_k))/\tau)]$$

$$\mathbf{v}_k^u = \gamma(\sum_f^F \frac{1}{\sqrt{F \times J}} \sum_j^J \mathbf{A}_{fjk}^u \mathbf{h}_{fj}^u) \ \forall k = 1, 2, ..., K \tag{3}$$

$w$ is a linear projection. $sim(\cdot, \cdot)$ is cosine similarity. $\tau_0, \tau$ are hyper-parameters. $\gamma$ is activation function. Each facet has the same weight $1/\sqrt{F \times J}$, modeling prior belief of uniform facet distribution.

$A_{fjk}^u$ is the binding score of low-level interest $j$ under facet $f$ to high-level interest $k$, consisting of the softmax over $K$ prototypes and the softmax over $J$ clusters. *Firstly*, as output of softmax over $K$ prototypes are non-negative and sums to 1, this requires $K$ prototypes to compete to attend low-level interests from $\mathbf{H}^u$. Thus, it encourages high-level interests to be different from others to capture the diversity of user's preferences. *Secondly*, softmax over $J$ clusters under each facet $f$ creates another competition between $J$ clusters, which constraints each prototype to mainly bind to one item characteristic. This constraint enables wisely binding for more interpretable interests, e.g., *low top* and *high top* shoes are tied to different high-level interests as it is unnatural to combine these two features into one interest.

After obtaining high-level interest representation of $u$, we calculate an *item-interest score matrix* as

$$\mathbf{B}_{ik}^u = \underset{1,2,...,K}{softmax}(\frac{\sum_{f,j} \mathbf{C}_{fij} \times \mathbf{A}_{fjk}^u}{\sqrt{F \times J}}), \mathbf{B}^u = \{\mathbf{B}_{ik}^u\}_{i=1,k=1}^{N,K} \in \mathbb{R}^{N \times K} \tag{4}$$

Until now we have $\mathbf{C}$ showing us the relations between items and clusters and $\mathbf{A}^u$ capturing the relations between clusters and user $u$'s interests. Therefore, $\mathbf{B}^u$ in Equation 4 describes the relations between items and user $u$'s interests, which is used in prediction step in Equation 6.

## 4.3 MODEL LEARNING

**Micro disentanglement.** We follow the common practice in VAE literature Higgins et al. (2017); Ma et al. (2019) to derive micro-disentanglement, i.e., disentangling dimensions of an interest representation. Firstly, $\mathbf{v}_k^u$ in Equation 3 is processed by *encoder* $g^0 : \mathbb{R}^{d^{enc}} \to \mathbb{R}^{2d}$ to estimate the parameters of variational distribution $\boldsymbol{\mu}_k^u \in \mathbb{R}^d$ and $\boldsymbol{\sigma}_k^u \in \mathbb{R}^d$ as following

$$(\mathbf{a}_k^u, \mathbf{b}_k^u) = g^0(\mathbf{v}_k^u) \implies \boldsymbol{\mu}_k^u = \mathbf{a}_k^u/||\mathbf{a}_k^u||_2; \quad \boldsymbol{\sigma}_k^u = \sigma_0 \cdot exp(-\frac{1}{2}\mathbf{b}_k^u) \tag{5}$$

$\sigma_0$'s value is around 0.1 Ma et al. (2019). The *final representation* of $k^{th}$ interest of user $u$, i.e., $\mathbf{z}_k^u \in \mathbb{R}^d$, is sampled from Gaussian distribution with estimated parameters, i.e., $\mathbf{z}_k^u \sim \mathcal{N}(\boldsymbol{\mu}_k^u, [diag(\boldsymbol{\sigma}_k^u)]^2) \ \forall k = 1, 2, ..., K$. A *regularization term* based on Kullback-Leibler (KL) divergence is added to match the estimated variational distribution with prior distribution, i.e., $D_{KL}(q(\mathbf{z}^u|\mathbf{x}^u, \mathbf{C})||p(\mathbf{z}^u))$. In which, $q(\mathbf{z}^u|\mathbf{x}^u, \mathbf{C}) = \prod_{k=1}^K q(\mathbf{z}_k^u|\mathbf{x}^u, \mathbf{C}) = \prod_{k=1}^K \mathcal{N}(\boldsymbol{\mu}_k^u, [diag(\boldsymbol{\sigma}_k^u)]^2)$ is variational distribution and $p(\mathbf{z}^u) = \mathcal{N}(\mathbf{0}, (\sigma_0)^2\mathbf{I})$ is factorized prior distribution to achieve micro-disentanglement. $D_{KL}(q||p) \to 0$ when $q$ and $p$ matches.

**Decoder.** Given $\{\mathbf{z}_k^u\}_{k=1}^K$, *decoder* predicts the probability that user $u$ interacts with item $i$ as

$$p(y_i^u) = \frac{\sum_{k=1}^K \mathbf{B}_{ik}^u r(\mathbf{z}_k^u)}{\sum_{i=1}^N \sum_{k=1}^K \mathbf{B}_{ik}^u r(\mathbf{z}_k^u)} \quad \text{with} \quad r(\mathbf{z}_k^u) = exp(sim(\mathbf{z}_k^u, \mathbf{T}_i)/\tau_{dec}) \tag{6}$$

$\mathbf{T}_i$ is item $i$'s vector. $sim(\cdot, \cdot)$ is cosine similarity. $\tau_{dec}$ is the temperature hyper-parameter. $p(y_i^u)$ is normalized over $N$ items. $\mathbf{B}_{ik}^u$ from Equation 4 is used to weight the prediction of an item, i.e., higher weight is given to an item provided its higher level of similarity with current interest.

**Learning objective.** FACETVAE minimizes an objective summing over a batch of user $\mathcal{B}^{user}$

$$\mathcal{L} = \sum_{u \in \mathcal{B}^{user}} [\mathcal{L}_{recon}^u + \mathcal{L}_{reg}^u] = \sum_{u \in \mathcal{B}^{user}} [\sum_{i=1}^N -y_i^u ln(p(y_i^u)) + \beta D_{KL}(q(\mathbf{z}^u|\mathbf{x}^u, \mathbf{C})||p(\mathbf{z}^u))] \tag{7}$$

For each user, $\mathcal{L}_{recon}^u$ is to reconstruct the observed interactions of user $u$ via cross-entropy loss as $p(y_i^u)$ follows categorical distribution. $\mathcal{L}_{reg}^u$ is a regularization term as described in Section 4.3. A hyper-parameter $\beta$ is introduced to control the influence of regularization objective versus the recommendation objective, similar to Multi-VAE Liang et al. (2018) and MacridVAE Ma et al. (2019).

## 5 EXPERIMENTS

**Datasets.** We consider three real-world datasets: i) **MovieLens-1M (ML-1M)**[2] (6,035 users, 3,126 movies, 574,376 ratings); ii) **CiteULike-a**[3] (5,551 users, 16,945 ariticles, 204,929 interactions); iii) **Yelp**[4] (29,111 users, 22,121 businesses, 1,052,627 reviews). We regard movies/articles/businesses as items and ratings/reviews as interactions. The implementation can be found in the link `https://github.com/PreferredAI/FacetVAE`.

**Competitors.** We compare FACETVAE against closely related baselines, i.e., disentangled representation **MacridVAE** Ma et al. (2019), **DGCF** Wang et al. (2020), **DCCF** Ren et al. (2023); multi-vector user representation **DPCML** Bao et al. (2022); VAE-based **RecVAE** Shenbin et al. (2020). We also include recently developed Collaborative Filtering models, i.e., contrastive learning **NCL** Lin et al. (2022), **SimGCL** Yu et al. (2022), representation theory-based **DirectAU** Wang et al. (2022b), graph-based **UltraGCN** Mao et al. (2021b), cosine-contrastive loss-based **SimpleX** Mao et al. (2021a).

**Hyper-parameter settings.** To ensure fair comparison, we tune the hyper-parameters of baselines following original papers. Dimension $d = 64$ and the number of user interests $K = 4$, where applicable, are fixed for all models. The details of hyper-parameter tuning is presented in supplementary materials.

**Metrics.** We report full ranking evaluation Zhao et al. (2020) of Recall and Normalized Discounted Cumulative Gain (NDCG) Tamm et al. (2021) as recommendation metrics. Both metrics are truncated at top 20 and top 50. Recall and NDCG are abbreviated as R and N in tables, respectively.

### 5.1 RECOMMENDATION PERFORMANCE COMPARISON

Table 1: Recommendation performance comparison. Boldfaced numbers are the highest while the runners-up are underlined. $\diamond$ denotes statistical significance between the boldfaced and the underlined on a paired t-test with p-value $< 0.01$. R and N stand for Recall and NDCG, respectively.

| Model | CiteULike-a | | | | Yelp | | | | ML-1M | | | |
|---|---|---|---|---|---|---|---|---|---|---|---|---|
| | R@20 | R@50 | N@20 | N@50 | R@20 | R@50 | N@20 | N@50 | R@20 | R@50 | N@20 | N@50 |
| SimpleX | 0.2607 | 0.3857 | 0.1534 | 0.1865 | 0.1163 | 0.2116 | 0.0611 | 0.0857 | 0.2706 | 0.4262 | 0.1988 | 0.2486 |
| NCL | 0.2378 | 0.3654 | 0.1353 | 0.1688 | 0.1551 | 0.2624 | 0.0851 | 0.1130 | 0.2766 | 0.4409 | 0.2026 | 0.2557 |
| DirectAU | 0.2534 | 0.3842 | 0.1412 | 0.1751 | 0.1677 | 0.2763 | 0.0994 | 0.1274 | 0.2695 | 0.4209 | 0.1947 | 0.2428 |
| SimGCL | 0.2444 | 0.3706 | 0.1375 | 0.1703 | 0.1634 | 0.2711 | 0.0929 | 0.1207 | 0.2870 | 0.4537 | 0.2082 | 0.2623 |
| UltraGCN | 0.2575 | 0.3839 | 0.1497 | 0.1835 | 0.1114 | 0.2065 | 0.0576 | 0.0820 | 0.2836 | 0.4467 | 0.2079 | 0.2603 |
| DGCF | 0.2066 | 0.3348 | 0.1165 | 0.1499 | 0.1337 | 0.2375 | 0.0713 | 0.0981 | 0.2598 | 0.4228 | 0.1890 | 0.2414 |
| DCCF | 0.2210 | 0.3536 | 0.1245 | 0.1589 | 0.1537 | 0.2573 | 0.0881 | 0.1150 | 0.2694 | 0.4360 | 0.1990 | 0.2523 |
| DPCML | 0.2498 | 0.3770 | 0.1429 | 0.1759 | 0.1125 | 0.2094 | 0.0576 | 0.0824 | 0.2754 | 0.4319 | 0.2050 | 0.2553 |
| RecVAE | 0.2398 | 0.3481 | 0.1435 | 0.1725 | 0.1137 | 0.2104 | 0.0596 | 0.0844 | 0.2952 | **0.4607** | 0.2146 | 0.2678 |
| MacridVAE | 0.2744 | 0.3974 | 0.1632 | 0.1958 | 0.1794 | 0.2835 | 0.1134 | 0.1405 | 0.2925 | 0.4553 | 0.2138 | 0.2669 |
| FACETVAE | **0.2837**$^\diamond$ | **0.4100**$^\diamond$ | **0.1695**$^\diamond$ | **0.2029**$^\diamond$ | **0.1859**$^\diamond$ | **0.2901**$^\diamond$ | **0.1192**$^\diamond$ | **0.1463**$^\diamond$ | **0.2968** | 0.4586 | **0.2178**$^\diamond$ | **0.2707**$^\diamond$ |

Table 1 reports the recommendation accuracy of FACETVAE and baselines. The results show that FACETVAE achieves significantly higher accuracy than those of baselines on CiteULike-a and Yelp. On ML-1M, FACETVAE demonstrates better performance than RecVAE, the best baseline, w.r.t. 3 out of 4 metrics. Notably, FACETVAE only requires roughly $1/3$ of RecVAE's number of parameters ($\sim 1.2$M vs. $\sim 3.6$M) to get the reported results, demonstrating the efficiency of FACETVAE.

There are two key takeaways. *First*, VAE-based models are top performing on all chosen datasets, demonstrating the strength of VAE framework that FACETVAE inherits. For example, on CiteULike-a and Yelp, MacridVAE achieves much better performance than both disentangled/multi-interest modeling models (DGCF, DPCML, DCCF) and single representation models (SimpleX, DirectAU, SimGCL, *inter alia*). On ML-1M, despite modeling user interest as a single vector, RecVAE outperforms DGCF, DCCF and DPCML which disentangle multiple factors of user interests. *Second*, prototype-based representation learning plays the key role in modeling multiple user interests under VAE framework. This is evidenced by the markedly performance gap between MacridVAE (using prototype-based representation) and RecVAE (without prototype-based representation) on CiteULike-a and Yelp. FACETVAE

---

[2]https://grouplens.org/datasets/movielens/

[3]http://wanghao.in/CDL.htm

[4]https://www.yelp.com/dataset

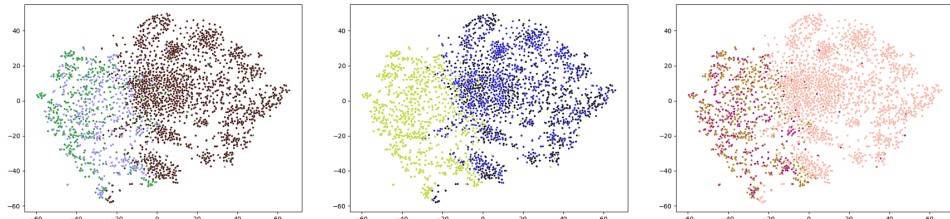

Figure 3: Item space structures produced by FACETVAE on ML-1M. Each plot shows multiple clusters of a facet. Items are colored according to $\arg\max_j \mathbf{C}_{fij}$ in Equation 1. It can be seen that *green* and *purple* groups of the first facet (the first plot) are grouped into one cluster (*genoa lemon*) in the second facet (the second plot). Contrarily, the *brown* group of the first facet is divided into *black* and *blue* groups in the second facet. In the third facet, these two are clustered into *orange* group.

generalizes MacridVAE via disentangling prototype-based representation under multi-faceted lens, achieving higher accuracy than both MacridVAE and RecVAE on three datasets.

## 5.2 MODEL STUDIES

**Efficiency Analysis.** FACETVAE and the closest baseline MacridVAE are bounded by the computational cost of grouping $N$ items into clusters. While it is $\mathcal{O}(KN)$ complexity in MacridVAE given $K$ clusters, that of FACETVAE is $\mathcal{O}(FJN + FJK)$. In which $\mathcal{O}(FJN)$ is the complexity of grouping items under $F$ facets each has $J$ clusters and $\mathcal{O}(FJK)$ is the complexity of binding block. Despite requiring higher computational demand, i.e., $F \times J$ is larger than $K$, FACETVAE's complexity is still a linear function of number of items $N$ as binding block's complexity *does not* depend on $N$. More importantly, FACETVAE achieves significantly higher accuracy than MacridVAE thanks to multi-faceted item grouping. To verify, we report the running time of FACETVAE and MacridVAE on Yelp dataset. On the other datasets, the running time of two models are roughly the same. For MacridVAE, K is 16 as it produces the best results. For FACETVAE, K is 8 achieving higher Recall and comparable NDCG compared to MacridVAE (as presented in supplementary materials). Then, the training time (second/epoch) and inference time (second) of FACETVAE are 10.068s and 2.311s, respectively. Those of MacridVAE are 9.736s and 1.963s, respectively. Clearly, FACETVAE only requires slightly higher running time than MacridVAE yet achieves better overall performance.

Table 2: Multi-faceted item space disentangling. Given $F \times J = 12$ prototypes, the total number of item characteristics is $J^F$. Boldfaced number is the best while runner-up is underlined per column.

| Setting | | ML-1M | | | | CiteULike-a | | | | Yelp | | | |
|---|---|---|---|---|---|---|---|---|---|---|---|---|---|
| F | J | R@20 | R@50 | N@20 | N@50 | R@20 | R@50 | N@20 | N@50 | R@20 | R@50 | N@20 | N@50 |
| 1 | 12 | 0.2936 | 0.4577 | 0.2110 | 0.2646 | **0.2824** | 0.4069 | **0.1692** | 0.2022 | 0.1804 | 0.2857 | 0.1146 | 0.1419 |
| 2 | 6 | 0.2951 | **0.4579** | 0.2131 | 0.2663 | 0.2819 | 0.4077 | 0.1688 | 0.2021 | 0.1848 | **0.2912** | 0.1180 | 0.1456 |
| 3 | 4 | **0.2968** | 0.4575 | 0.2168 | 0.2693 | 0.2820 | 0.4076 | 0.1690 | 0.2023 | **0.1856** | **0.2912** | 0.1189 | 0.1463 |
| 4 | 3 | 0.2964 | 0.4570 | **0.2173** | **0.2697** | 0.2806 | **0.4086** | 0.1688 | **0.2026** | 0.1850 | 0.2885 | **0.1200** | **0.1470** |
| 6 | 2 | 0.2934 | 0.4546 | 0.2152 | 0.2676 | 0.2778 | 0.4040 | 0.1661 | 0.1997 | 0.1820 | 0.2831 | 0.1191 | 0.1455 |

**Multi-faceted item space disentangling.** We fix the total number of prototypes used to group items is 12 then we vary $F$ and $J$ satisfying $F \times J = 12$. When $F = 1$, it reduces to single-faceted item grouping. Table 2 presents the results. The key takeaways are *first*, grouping item space under multiple facets, i.e., $F > 1$, results in overall higher recommendation accuracy, demonstrating its ability to discover fine-grained structure of item space. *Second*, setting $F$ and $J$ of which $J^F$ is large, e.g., $F = 3, J = 4$ or $F = 4, J = 3$, generally results in better performance. These results are consistent with our hypothesis in Section 4.1 that the number of item characteristics FACETVAE can discover is up to $J^F$. The more item characteristics are discovered, the better modeling user interest is.

**Item space disentangling visualization.** We visualize the item groups produced by FACETVAE in Figure 3 to qualitatively examine whether FACETVAE can discover multi-faceted item space structure. We use t-SNE van der Maaten & Hinton (2008) to visualize item representations on

2D space. Evidently, the neighbors of items, i.e., those with the same color, across facets vary, demonstrating that FACETVAE is capable of disentangling multi-faceted item space.

Table 3: Analysis of low-level and high-level user's interests. F = J = 3 for CiteULike-a and ML-1M while F = J = 4 for Yelp. Boldfaced number is the highest in each column. (*) refers to Equation 3.

| User Interest | Binding block setting | Number of user interests | ML-1M | | CiteULike-a | | Yelp | |
|---|---|---|---|---|---|---|---|---|
| | | | R@20 | N@20 | R@20 | N@20 | R@20 | N@20 |
| low level | removed | $F \times J$ | 0.2826 | 0.2091 | 0.2710 | 0.1628 | 0.1835 | 0.1242 |
| high level | softmax over K prototypes only (*) | 4 | 0.2962 | 0.2148 | 0.2685 | 0.1590 | 0.1818 | 0.1159 |
| | softmax over J clusters only (*) | 4 | 0.2908 | 0.2133 | 0.2820 | 0.1692 | 0.1794 | **0.1249** |
| | bi-directional | 4 | **0.2968** | **0.2178** | **0.2837** | 0.1695 | **0.1859** | 0.1192 |
| | bi-directional | 8 | 0.2948 | 0.2171 | 0.2837 | **0.1696** | 0.1846 | 0.1214 |

**Low-level vs. high-level user interests.** Table 3 presents results when using low-level and high-level user interests (Section 4.2) for recommendation. Clearly, leveraging high-level user interests results in higher accuracy as they are more expressive than low-level ones. For instance, using 4 high-level user interests leads to much better accuracy than using 9 low-level user interests on CiteULike-a and ML-1M. On Yelp, we observe the same trend yet it depends on the setting of binding block.

**Analysis of binding block.** We also study binding block via reported results in Table 3. *Firstly*, bi-directional binding obviously achieves larger performance than uni-directional counterpart, i.e., performing softmax over $K$ prototypes or $J$ clusters only. *Secondly*, bi-directional binding's effect is data-dependent. While softmax over $K$ prototypes has stronger influence on ML-1M than softmax over $J$ clusters, we observe the opposite trend on CiteULike-a. On Yelp, this effect is metric-dependent.

**Interpretability of user's interests.** We study the interpretability of user's interests produced by FACETVAE. After training, we retrieve three items with highest score predicted by each interest of a user (see Equation 6) in Table 4. These examples suggest that FACETVAE has the potential to discover the multiple interpretable interests of users. However, we note that user's interests are derived in an *unsupervised* manner, which may result in one interest with many items dominates the less popular ones or ambiguous interests.

Due to limited space, we present *more experimental results* to understand FACETVAE, including analysis of number of user interests $K$, values of $F$ and $J$ versus recommendation performance and the influence of micro-disentanglement on recommendation accuracy, in supplementary materials.

## 6 CONCLUSION

We introduce FACETVAE to resolve shortcomings of VAE-based disentangled recommendation models, including inadequately item space discovering, same level of granularity between user interests and item space assumption, which causes a dilemma and improperly user interest complexity handling. FACETVAE is characterized by three main innovations *1)* disentangling item space under multi-faceted manner, *2)* binding compositional user interests from low-level ones discovered from item space and *3)* effectively binding user interests via bi-directional binding block. Future work extending FACETVAE includes improving the efficiency of multi-faceted item grouping and discovering the number of facets and the number of clusters per facet in a data-driven manner.

Table 4: Top three items (movies) with highest score predicted by multiple interests of a user on ML-1M. We present movie's title and tags (inside parentheses). Three interests have their own semantics, i.e., *Star Wars series*, *comedy movies*, and *drama movies*, respectively.

| Interest 1 | Interest 2 | Interest 3 |
|---|---|---|
| 1. Star Wars: Episode V - The Empire Strikes Back (Action \| Adventure \| Drama \| Sci-Fi \| War) | 1. High Fidelity (Comedy) | 1. American Beauty (Comedy \| Drama) |
| 2. Star Wars: Episode VI - Return of the Jedi (Action \| Adventure \| Drama \| Sci-Fi \| War) | 2. Wonder Boys (Comedy \| Drama) | 2. Braveheart (Action \| Drama \| War) |
| 3. Star Wars: Episode IV - A New Hope (Action \| Adventure \| Drama \| Sci-Fi \| War) | 3. American Beauty (Comedy \| Drama) | 3. The Shawshank Redemption (Drama) |

ACKNOWLEDGMENTS

This research/project is supported by the National Research Foundation, Singapore under its AI Singapore Programme (AISG Award No: AISG2-RP-2021-020).

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

# A  Supplementary Materials

## A.1  Datasets

We conduct experiments on three real-world datasets. **MovieLens-1M (ML-1M)**[5] (6,035 users, 3,126 items, 574,376 interactions) contains user's ratings for movies. A review that a user wrote for an item is considered an interaction between the two. **CiteULike-a**[6] (5,551 users, 16,945 items, 204,929 interactions) contains academic articles and users' log history. An interaction between a user and an item (article) is a user saving an article to their collection. **Yelp**[7] (29,111 users, 22,121 items, 1,052,627 interactions) contains reviews (interactions) written by users for businesses (items). Due to its huge volume, we only consider interactions from 2016 onwards. We pre-process ML-1M following Ma et al. (2019) and Yelp following Lin et al. (2022) and keep CiteULike-a dataset as the original. We construct training, validation and test sets by randomly dividing users' interactions with ratio 8:1:1, respectively. All cold-starts users and items in validation and test sets are discarded.

## A.2  Implementation Details

Algorithm 1 presents detailed training procedure of FACETVAE.

The code, data and related materials can be found in the link `https://github.com/PreferredAI/FacetVAE`

## A.3  Experiments

### A.3.1  Details on Comparative Methods

- **MacridVAE** Ma et al. (2019) disentangles macro and micro levels of user's interests based on $\beta$-VAE.
- **DGCF** Wang et al. (2020) iteratively refines multiple factor representations of users and items to disentangle their multiple interests from interaction graph.
- **DPCML** Bao et al. (2022) improves Collaborative Metric Learning by using multiple representations for users.
- **RecVAE** Shenbin et al. (2020) improves VAE-based CF model by presenting novel composite prior distribution, a new method to set $\beta$ in $\beta$-VAE and an alternative training algorithm.
- **DirectAU** Wang et al. (2022b) improves representation learning in CF by optimizing uniformity and alignment.
- **NCL** Lin et al. (2022) enhances graph-based CF by incorporating structural and semantic neighbors via contrastive learning.
- **SimpleX** Mao et al. (2021a) improves collaborative filtering by designing novel cosine contrastive loss and incorporating large negative sampling.
- **SimGCL** Yu et al. (2022) proposes a novel contrastive learning approach for CF by augmenting random noise to representations and regulating their uniformity.
- **UltraGCN** Mao et al. (2021b) approximates message passing limit of message passing layer and leverage item-item relation information to improves graph-based CF.
- **DCCF** Ren et al. (2023) leverages an adaptive approach for self-supervised augmentation to disentangles intents behind user-item interactions.

### A.3.2  Hyper-parameter Settings

For FACETVAE, we tune the hyper-parameters in the same range as MacridVAE. Dropout rate is 0.5. $\sigma_0 \in \{0.05, 0.075, 0.1\}$. $\beta = min(\beta_0, \frac{update}{T})$, with $\beta_0 \in \{0.01, 0.05, 0.1, 0.2, 0.5, 1\}$ and $update$ is the number of model's parameters updates, $T \in \{0.1k, 0.5k, 1k, 5k, 10k, 20k\}$. $\tau \in [0.1, 0.2]$, $\tau_{dec} \in \{0.1, 0.15, 0.2\}$, $\tau_0 \in \{0.1, 0.2, 0.5, 1, 5, 8\}$. $\gamma$ is $tanh$ while $\gamma_0$ is LeakyReLU(0.3) for

---

[5]https://grouplens.org/datasets/movielens/

[6]http://wanghao.in/CDL.htm

[7]https://www.yelp.com/dataset

---

**Algorithm 1:** Training procedure of FACETVAE

---

**Input:**

- User interacted items $\mathbf{x}^u = \{i : y_i^u = 1\}$
- Model parameters $\Theta$
    - item matrix in decoder $\mathbf{T} \in \mathbb{R}^{N \times d}$, context matrix $\mathbf{E} \in \mathbb{R}^{N \times d^{enc}}$ and bias vector $\mathbf{b}^{enc}$
    - clustering prototype representations $\mathbf{P} \in \mathbb{R}^{F \times J \times d}$ and binding prototype representations $\mathbf{Q} \in \mathbb{R}^{K \times d}$
    - parameters of linear projection layer $w : \mathbb{R}^d \to \mathbb{R}^{d^{enc}}$ and neural network $g^0 : \mathbb{R}^{d^{enc}} \to \mathbb{R}^{2d}$
- Number of facets $F$, number of clusters per facet $J$, number of interests $K$

**Output:**

- Updated $\Theta$

1 **Function** *Multi-faceted item grouping (T, P)*
2    **for** *f = 1 to F* **do**
3      **for** *i = 1 to N* **do**
4        $\mathbf{C}_{fij} = Gumbel - Softmax([\mathbf{s}_{fi1}, \mathbf{s}_{fi2}, ..., \mathbf{s}_{fiJ}])$   with   $\mathbf{s}_{fij} =$
       $(\mathbf{T}_i)^T \mathbf{P}_{fj}/(\tau \cdot ||\mathbf{T}_i||_2 \cdot ||\mathbf{P}_{fj}||_2)$   $\forall j = 1, 2, ..., J$
5    **return** $\mathbf{C} \in \mathbb{R}^{F \times N \times J}$
6 **Function** *Low-level interest aggregation($\mathbf{x}^u, \mathbf{C}$)*
7    **for** *f = 1 to F* **do**
8      **for** *j = 1 to J* **do**
9        $\mathbf{h}_{fj}^u = \gamma_0 \left( \frac{\sum_{i \in \mathbf{x}^u} \mathbf{C}_{fij} \cdot \mathbf{E}_i}{\sqrt{Z}} + \mathbf{b}^{enc} \right)$   with   $Z = \sum_{i \in \mathbf{x}^u} (\mathbf{C}_{fij})^2$ // $\gamma_0$ – non-linear
       activation function
10    **return** $\mathbf{H}^u = \{\mathbf{h}_{fj}^u\} \forall f = 1, 2, ..., F; \forall j = 1, 2, ..., J, \mathbf{H}^u \in \mathbb{R}^{B \times F \times J \times d^{enc}}$ // $B$ – batch
       size
11 **Function** *Binding block($\mathbf{H}^u = \{\mathbf{h}_{fj}^u\}_{f=1,j=1}^{F,J}$)*
12    **for** *k = 1 to K* **do**
13      **for** *f = 1 to F* **do**
14        **for** *j = 1 to J* **do**
15          $\mathbf{A}_{fjk}^u = \frac{1}{2}[\text{softmax}_K(sim(\mathbf{h}_{fj}^u, w(\mathbf{Q}_k))/\tau_0) + \text{softmax}_J(sim(\mathbf{h}_{fj}^u, w(\mathbf{Q}_k))/\tau)]$
         // $sim(\cdot, \cdot)$ is cosine similarity
16      $\mathbf{v}_k^u = \gamma(\sum_f^F \frac{1}{\sqrt{F \times J}} \sum_j^J \mathbf{A}_{fjk}^u \mathbf{h}_{fj}^u)$ // $\gamma$ – non-linear activation function
17      $\mathbf{B}_{ik}^u = \text{softmax}_K \left( \frac{\sum_{f,j} \mathbf{C}_{fij} \times \mathbf{A}_{fjk}^u}{\sqrt{F \times J}} \right) \forall i = 1, 2, ..., N$
18    **return** $\mathbf{V}^u = \{\mathbf{v}_k^u\}_{k=1}^K \in \mathbb{R}^{B \times K \times d^{enc}}; \mathbf{B}^u \in \mathbb{R}^{B \times N \times K}$ // $B$ – batch size
19 **Function** *Encoder($\mathbf{V}^u = \{\mathbf{v}_k^u\}_{k=1}^K$)*
20    **for** *k = 1 to K* **do**
21      $(\mathbf{a}_k^u, \mathbf{b}_k^u) = g^0(\mathbf{v}_k^u)$
22      $\boldsymbol{\mu}_k^u = \mathbf{a}_k^u / ||\mathbf{a}_k^u||_2$
23      $\boldsymbol{\sigma}_k^u = \sigma_0 \cdot exp(-\frac{1}{2}\mathbf{b}_k^u)$
24      $\mathbf{z}_k^u \sim \mathcal{N}(\boldsymbol{\mu}_k^u, [diag(\boldsymbol{\sigma}_k^u)]^2)$
25    **return** $\{\mathbf{z}_k^u\}_{k=1}^K \in \mathbb{R}^{B \times K \times d}$
26 **Function** *Decoder ($\{\mathbf{z}_k^u\}_{k=1}^K, \mathbf{B}^u$)*
27    **for** *k = 1 to K* **do**
28      $r(\mathbf{z}_k^u) = exp(sim(\mathbf{z}_k^u, \mathbf{T}_i)/\tau_{dec})$ // $sim(\cdot, \cdot)$ is cosine similarity
29    $p(y_i^u) = \frac{\sum_{k=1}^K \mathbf{B}_{ik}^u r(\mathbf{z}_k^u)}{\sum_{i=1}^N \sum_{k=1}^K \mathbf{B}_{ik}^u r(\mathbf{z}_k^u)} \forall i = 1, 2, ..., N$
30    **return** $\{p(y_i^u)\}_{i=1}^N$
31 $\mathbf{C} \leftarrow Multi - faceted\ item\ grouping\ (\mathbf{T}, \mathbf{P})$
32 $\mathbf{H}^u \leftarrow Low - level\ interest\ aggregation(\mathbf{x}^u, \mathbf{C})$
33 $\mathbf{V}^u, \mathbf{B}^u \leftarrow Binding\ block(\mathbf{H}^u)$
34 $\{\mathbf{z}_k^u\}_{k=1}^K \leftarrow Encoder(\mathbf{V}^u)$
35 $\{p(y_i^u)\}_{i=1}^N \leftarrow Decoder(\{\mathbf{z}_k^u\}_{k=1}^K, \mathbf{B}^u)$
36 Calculate loss $\mathcal{L} = \sum_{u \in \mathcal{B}^{user}} [\sum_{i=1}^N -y_i^u ln(p(y_i^u)) + \beta D_{KL}(q(\mathbf{z}^u | \mathbf{x}^u, \mathbf{C}) || p(\mathbf{z}^u))]$
37 Update $\Theta$ to minimize $\mathcal{L}$

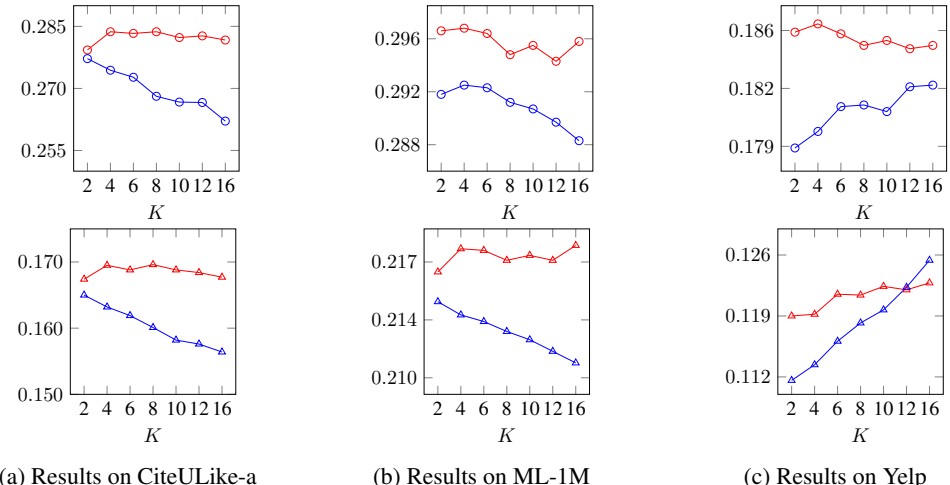

(a) Results on CiteULike-a     (b) Results on ML-1M     (c) Results on Yelp

Figure 4: Performance of FACETVAE (red line) w.r.t. number of interests $K$. We include results of MacridVAE (blue line) for contrasting. Small circles are Recall@20 while triangles are NDCG@20.

ML-1M and $tanh$ for other datasets. $d^{enc} = d = 64$ for CiteULike-a and Yelp and $d^{enc} = 300$ for ML-1M. $F = J = 3$ for CiteULike-a and ML-1M and $F = J = 4$ for Yelp. All models are trained on NVIDIA RTX 2080 Ti GPU machine ten times with different random seeds. For FACETVAE, we set maximal number of epochs is 200 and stop training after 15 epochs without improving Recall@20 on validation set. Averaged results over ten runs on test set are reported.

### A.3.3 Additional Results

**Number of user's interests $K$.** We analyze performance of FACETVAE and MacridVAE w.r.t. various numbers of user's interests $K$ in Figure 4. There are three key takeaways. *First*, FACETVAE achieves higher recommendation accuracy than MacridVAE across various values of $K$ w.r.t. most of the metrics, except NDCG@20 with $K = 16$. These evidences ascertain that multi-faceted item space disentangling adopted in FACETVAE can discover more useful item characteristics for modeling user interests. *Second*, FACETVAE is robust to the number of user interests, which is evidenced in Figure 4 (a) and (b), i.e., FACETVAE's performance does not get hurted when increasing the number of user interests. Contrarily, MacridVAE assumes the number of user's interests equals to the number of item groups, which causes representation dilemma, i.e., a small number of item groups might be sufficient to capture user's interests yet insufficient to model item space structure, while a large number of item groups might be sufficient to describe item space but might cause redundant and noisy user interests. This dilemma negatively affects model performance, which is shown in Figure 4 (a) and (b) on CiteULike-a and ML-1M, respectively. *Third*, FACETVAE demonstrates its efficiency when dealing with large dataset. It is clear in Figure 4 (c) that FACETVAE with $K = 4$ achieves higher Recall@20 than MacridVAE with $K = 10$, $K = 12$ or $K = 16$ on the largest dataset Yelp. A similar trend is also observed on NDCG@20 when setting $K = 6$ or $K = 8$ for MacridVAE and $K = 4$ for FACETVAE. It is note that setting a large value of $K$ is prohibitively expensive for huge datasets when predicting interactions between users and million of items as the complexity scales with $K$.

**Analysis of $F$ and $J$.** We investigate model performance w.r.t. various values of the number of facets $F$ and the number of item groups under each facet $J$ in Figure 5. The key observations are *first*, the values of $F$ and $J$ are data-dependent, e.g., on CiteULike-a and ML-1M, setting $F = 3$ and $J = 3$ achieves highest results while on Yelp Recall favors $F = 3$ and $J = 4$ or $J = 5$ and NDCG prefers $F = 5$ and $J = 4$. *Second*, small datasets prefer small values of $F$ and $J$ while large dataset favors large values of these two. This observation is expected as large data is usually more complicated in terms of item space structure. *Last but not least*, large values of $F$ and $J$ might result in redundant and noisy discovered item characteristics, which negatively affecting model performance.

**Micro-disentanglement vs. Recommendation Accuracy.** As described in the main text, $\beta$ controls the level of disentanglement between dimensions of representation vector. The value of $\beta$ follows

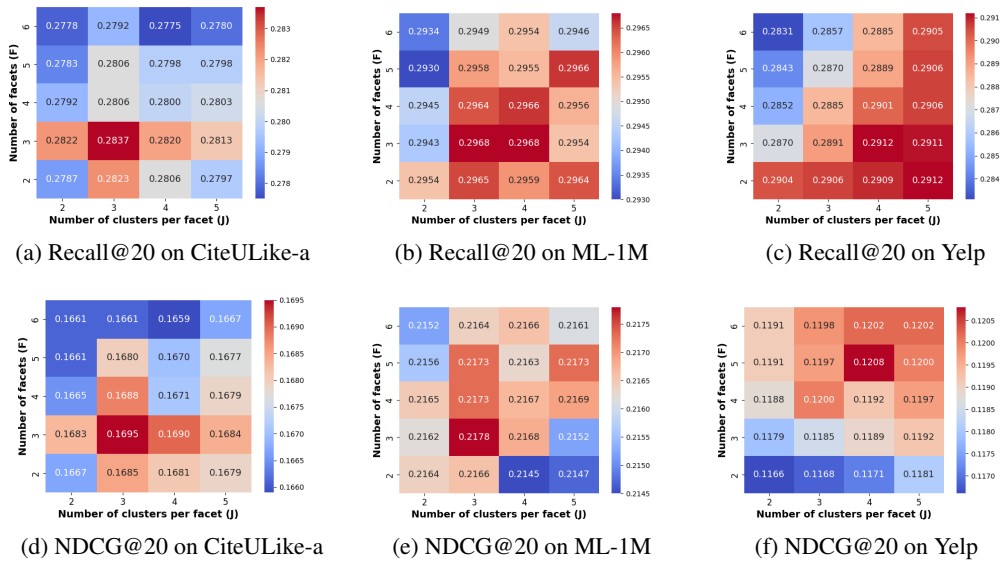

Figure 5: FACETVAE's performance w.r.t. $F$ and $J$.

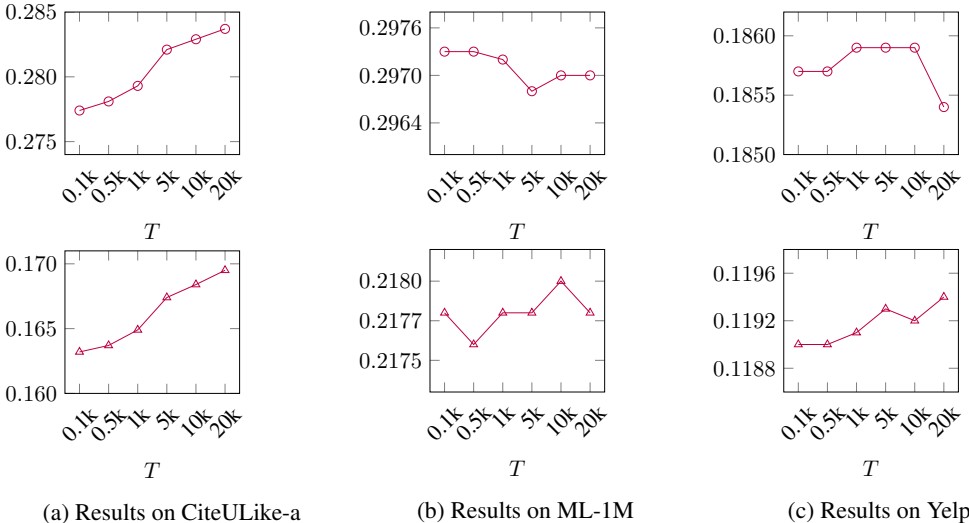

Figure 6: Performance w.r.t. number of annealing steps $T$. The higher $T$ is, the lower level of disentanglement is. Circle symbols are Recall@20 while triangle symbols are NDCG@20.

an annealing procedure, i.e., $min(\beta_0, \frac{update}{T})$ as in Ma et al. (2019). $\beta_0$ is the maximal value of $\beta$, $update$ is the number of model's parameters update by gradient descent and $T$ is the total of annealing steps. We fix $\beta_0$ as default choice and vary the values of $T$. Clearly, increasing $T$ results in lower level of disentanglement. We present model performance w.r.t. $T$ in Figure 6. There are two key takeaways. *For one*, we observe that on CiteULike-a, increasing $T$ to lower disentanglement level leads to improvement in both Recall and NDCG. Contrarily, ML-1M and Yelp are not sensitive to micro-disentanglement. *For another*, there is a trade-off between Recall and NDCG w.r.t. disentanglement level on these datasets. In general, three chosen datasets prefer low level of disentanglement, $T > 1k$, to achieve good performance w.r.t. Recall and NDCG. *In conclusion*, these evidences imply that the strong performance of FACETVAE mainly comes from the multi-faceted item disentangling and binding compositional user interests, verifying the effectiveness of our approach in this paper.

