# OpenReview forum: "Learning Multi-Faceted Prototypical User Interests"
_ICLR.cc/2024/Conference — ICLR 2024 poster_

### Official Review · Reviewer_JT1U · 2023-10-29

**Soundness:** 3 good
**Presentation:** 3 good
**Contribution:** 2 fair
**Rating:** 5
**Confidence:** 3

**Summary:**

This paper proposes FACETVAE to align user interests with multi-faceted item characteristics. The paper puts forward low-level and high-level user interests and introduces a bi-directional binding block to learn compositional user interests. Rich experimental results demonstrate the  good performance of FACETVAE.

**Strengths:**

- This paper is well motivated and the idea of learning multi-faceted prototypes is interesting.

- The paper conducts comprehensive experiments that encompass various aspects such as prediction performance, visualizations, parameter studies, and case studies.

- The authors have made their code and dataset publicly available.

**Weaknesses:**

1. The proposed method is somehow incremental and is a combination of existing techniques. This work is based on an existing work that learns prototypical representation of items and users that seperate items into different groups. This work further incorporates disentangled representation learning to explore item representations of various aspects of items, which has been also considered in previous works[1].
The main difference appears to be  the combination of these various aspects.
2. Figure 2 is confusing because it takes explicit user-item interaction in an rating form. However, the paper is designed based on implicit user-item interaction. This misleads the reader on the output form and training strategy of the proposed method.
3. Would it be better to include item attribute information in prototype learning as auxiliary information? Without low-level attributes as input, the so-called low-level user interest about a facet might also be a composite feature of some facets. This could result in the difference in meaning between the two levels of interests not being clear.
4.  More qualitative analysis and case studies about on how learned latent representations corresponds to different aspects of characteristics may help to better evaluate the interpretability of the model. For example, more details on how the "latent interest" corresponds to explicit interests 1, 2, and 3 in Table 4 would be appreciated.

[1] Learning Disentangled Representations for Recommendation

**Questions:**

Please refer to the weaknesses

---

> ### Author Response · Authors · 2023-11-15
> **Responses to Reviewer JT1U**
>
> Dear Reviewer **JT1U**,
>
> We would like to thank you for spending time reviewing our paper. We appreciate your constructive comments. We address your concerns as follows
>
> **Q1** *The proposed method is somehow incremental and is a combination of existing techniques*
>
> **A1** Regarding the motivation, the proposed method distinguishes itself by two key aspects: discovering multi-faceted item space and modeling the composition of user interests. These motivations have not been considered in [1].
>
> In terms of technical aspects, we designed a novel model including a prototype-based multi-faceted item clustering mechanism, a binding block to model user interest compositionality and the dedicated bi-directional binding block. These components have been empirically shown to be important to achieve higher recommendation accuracy. In contrast, these components were not studied in [1]. Moreover, our model is more general and effective than MacridVAE [1].
>
> As MacridVAE serves as the base of recently disentangled recommendation models, our idea improving MacridVAE, thus, can be applied to a range of problems.
>
> *[1] Ma et al. Learning Disentangled Representations for Recommendation. NeurIPS 2019.*
>
> **Q2** *Figure 2 is confusing between explicit user’s rating and implicit user-item interactions*
>
> **A2** In Figure 2, the input is binary rating vector, where the numbers are IDs of items. Shaded circles are those that user interacted with while dashed circles are unobserved interaction items.
>
> We updated Figure 2 and added clearer notations to ease the understanding.
>
> **Q3** *Would it be better to include item attribute information in prototype learning as auxiliary information?*
>
> **A3** It is an interesting direction to incorporate item attributes into the proposed approach. With item attributes such as item categories, one can formulate a classification task to match the learned item cluster with ground-truth categories as classification labels.
>
> In this paper, we consider user-item interaction data for collaborative filtering task, which is the most popular method in recommender systems. As future work, once item attributes are provided, one can integrate them into our proposed approach as aforementioned.
>
> **Q4** *How does the "latent interest" correspond to explicit interests 1, 2, and 3 in Table 4?*
>
> **A4** Each user's latent interest, which is denoted by **z**$^u_k$ (Equation 5) in the main text, is expected to capture one aspect of user preferences. It is derived in an unsupervised manner without any prior information. Thus, to understand the meaning of these latent interests, we find the top 3 items that are closest in representation space with those interests by leveraging output of decoder, i.e., top 3 items with highest decoded score. This score is simply the cosine similarity between latent interest representation and item representations stored in decoder (as shown in Equation 7). Then we visualize the item side information of those items in Table 4 to understand each latent interest.

---

> ### Author Response · Authors · 2023-11-21
> **Looking forward to your feedback**
>
> Dear Reviewer JT1U,
>
> Thank you again for your time reviewing our paper.
>
> We just want to post a friendly reminder that whether our rebuttal has addressed your concerns.
>
> If you have any questions, please let us know. We are happy to discuss.
>
> Best,
>
> Authors.

---

### Official Review · Reviewer_BBhv · 2023-11-02

**Soundness:** 4 excellent
**Presentation:** 4 excellent
**Contribution:** 3 good
**Rating:** 8
**Confidence:** 3

**Summary:**

This paper presented a new approach to uncover latent user interests for recommender system. Specifically, it proposed FACETVAE which tries to align user interests with multi-facted item characteristics and prototype-based user representation. It tries to separate granularity of user preferences from that of item space. WIth a dedicated bi-directional binding block, it demonstrates better recommendation performance compared to other baseline VAE approaches.

**Strengths:**

1. The paper is well organized and written.
2. The paper is well motivated.
3. The paper proposed a mathematically solid approach and demonstrated its clear strength over baselines.

**Weaknesses:**

The proposed approach has worse complexity than baseline MacridVAE. For real-world recommendation problems with huge spaces of users and items, the approach seems less applicable.

**Questions:**

1. For Table 1, what is the number of clusters and the number of parameters for the baseline MacridVAE?

2. Is it possible to report the performance with respect to the number of parameters, and computation time across the baselines and the proposed approach? That seems to be able to give readers a better understanding of accuracy and efficiency.

3. In Table 2, why is the number of prototypes fixed to 12 instead of other numbers? Can the best F and J have different trends with different number of prototypes?

---

> ### Author Response · Authors · 2023-11-15
> **Responses to Reviewer BBhv**
>
> Dear Reviewer **BBhv**,
>
> We would like to thank you for your acceptance and your constructive comments. We address your concerns as follows
>
> **W1** *The proposed approach has worse complexity than MacridVAE*
>
> **A1** The proposed method outperforms MacridVAE on chosen datasets. Despite having slightly higher complexity than MacridVAE in item clustering step, the complexity is a linear function of the number of items. Moreover, thanks to the multi-faceted item grouping step, FacetVAE can handle large and complex datasets, e.g., with 8 facets and 5 clusters per facet, FacetVAE can describe a dataset including up to $5^8$ (~400k) characteristics. Thus, the proposed method can be applied to large scale datasets.
>
> **Q1** *In table 1, what is the number of clusters and the number parameters for MacridVAE?*
>
> **A1** In table 1, we set the number of clusters for MacridVAE is 4, which is also the number of user interests. The number of user interests (K) in FacetVAE is also 4 for fair comparison. In supplementary Figure 4, we varied the number of user interests. In these experiments, FacetVAE still achieves higher recommendation accuracy than MacridVAE.
>
> We configure the settings of MacridVAE and FacetVAE almost the same for fair comparison. The main difference between parameters of these two is multi-faceted clustering step.
>
> FacetVAE uses slightly more parameters than MacridVAE, including $F * J * d^{enc}$ parameters and parameters of $w$ in Equation 3. With small values of $F$ and $J$ (3 or 4) and $d^{enc}$ is not too big (64 or 300), the difference between parameters of MacridVAE
> and FacetVAE is small. Table below shows the relative comparison w.r.t. the number of parameters between MacridVAE and FacetVAE
>
> |             | Citeulike-a |     Yelp   | ML-1M  |
> |:---------:|:------------:|:---------:|:------:|
> | FacetVAE  |   100.22%   | 100.18% | 101.5% |
> | MacridVAE |    100%     |  100%   |  100%  |
>
> **Q2** *Report the performance with respect to the number of parameters, and computation time across the baselines and the proposed approach*
>
> **A2** In the answer to **Q1** above, we (relatively) compare the number of parameters of FacetVAE  and the closest baseline MacridVAE. It is shown that FacetVAE just uses a small number of parameters more than MacridVAE.
>
> In Section 5.2, we already provided the comparison between running time to train single epoch (the reported numbers are averaged over 10 runs) of FacetVAE and MacridVAE. In summary, FacetVAE requires slightly more time than MacridVAE, demonstrating its efficiency.
>
> **Q3** *In Table 2, why is the number of prototypes fixed to 12 instead of other numbers? Can the best F and J have different trends with different number of prototypes?*
>
> **A3** Due to limited space, in Table 2, we fix the number of prototypes as 12 for illustration. In supplementary Figure 5, we report the accuracy with various values of number of facets $F$ and number of clusters per facet $J$, which correspond to various values of number of prototypes.

---

> ### Author Response · Authors · 2023-11-21
> **Looking forward to your feedback**
>
> Dear Reviewer BBhv,
>
> May we know that whether your questions have been addressed in the rebuttal above?
>
> Please let us know if you have further questions. We are happy to discuss.
>
> Best Regards,
>
> Authors.

---

### Official Review · Reviewer_Ruv4 · 2023-11-09

**Soundness:** 3 good
**Presentation:** 3 good
**Contribution:** 3 good
**Rating:** 6
**Confidence:** 4

**Summary:**

This work propose to learn multi-faceted prototypical user interests based on VAE framework. The proposed method is technically sound, although it can be viewed as a natural extension based on MacridVAE, which limits the technical contribution somehow. The paper is well written and easy to follow, related work is well discussed and the extensive experiments and ablation studies verify the model performance.

**Strengths:**

1. This paper is well motivated to solve the multi-facet user interest modeling problem based on VAE framework. The proposed method is well motivated and presented.
2. This paper is well written and easy to follow, especially the related work and limitation discussion in preliminaries.
3. Extensive experiments are conducted, with state-of-the-art methods as baselines. The results demonstrate the improvement.

**Weaknesses:**

1. In real world recommender systems, different product categories will have different facets, in total, it can reach up to thousands of facets in the recommendation space, especially when we want to do whole website recommendation. It's unclear how this method can handle and scale to large scale product facets.
2. This work can be viewed as an incremental upon MacridVAE, which limits the technical novelty contribution.
3. The performance on large scale (millions of users and items) datasets is unclear.
4. How to leverage product knowledge/other knowledge source to automatically choose #facets should be discussed to make it applicable in real world applications.

**Questions:**

See above.

---

> ### Author Response · Authors · 2023-11-15
> **Responses to Reviewer Ruv4**
>
> Dear Reviewer **Ruv4**,
>
> We would like to thank you for spending time reviewing our paper. We appreciate your constructive comments. Your concerns are addressed as follows
>
> **W1** *How can this method handle and scale to large scale product facets, which can reach up to thousands of facets?*
>
> **A1** The proposed method can handle large scale datasets including thousands of facets thanks to the multi-faceted modeling. For example, a dataset contains *m* categories $C_1, C_2 …, C_m$; each includes $F_1, F_2, …, F_m$ facets, respectively. A naïve solution grouping items using single facet would require $F_1 * F_2 * … * F_m$ clusters, which is exponentially large and thus it might be feasible to handle huge datasets. However, the proposed method only requires $F_1 + F_2 + … + F_m$ clusters, which scales linearly. Concretely, if there are 6 categories, each including 4 facets, the total number of required clusters for a naïve solution is $4^6 = 4096$ clusters while the proposed method needs $4 * 6 = 24$ clusters to model item space structure.
>
> **W2** *This work can be viewed as an incremental upon MacridVAE*
>
> **A2** Our work is distinct from MacridVAE in three aspects: discovering multi-faceted item space, binding compositional user representations and bi-directional binding block to model compositional user representations. These ideas were not discussed in MacridVAE. Moreover, these aspects have been shown to achieve higher recommendation accuracy.
>
> It is worth noting that MacridVAE serves as the base of recently developed recommendation models. Our key ideas for improving MacridVAE, therefore, are applicable to a wide range of problems.
>
> **W3** *The performance on large scale (millions of users and items) datasets is unclear*
>
> **A3** Currently, we are unable to conduct experiments on such huge datasets due to limited computational resources, which we leave these experiments for future work. In any case, the methodology we propose would scale linearly with the number of users or items.
>
> **W4** *How to leverage product knowledge/other knowledge source to automatically choose #facets should be discussed*
>
> **A4** When product knowledge, e.g., item category, is available, one could use this information to supervise the item grouping in our proposed method, i.e., a classification task between the learned item group and the ground-truth item category as label.

---

> ### Author Response · Authors · 2023-11-21
> **Looking forward to your feedback**
>
> Dear Reviewer Ruv4,
>
> Given the rebuttal above, may we know that whether we have addressed your concerns about our paper?
>
> We are happy to discuss if you have further questions.
>
> Best Regards,
>
> Authors.

---

### Meta-Review · Area_Chair_tRzP · 2023-12-05

**Metareview:**

The paper presents an approach to solving the multi-faceted user interest modeling problem using a VAE framework. The method's clear strength over baselines is demonstrated through extensive experiments, and the paper is well-organized, clearly written, and thorough in discussing related work and limitations. The method, while interesting, is seen as incremental, building upon existing techniques and frameworks such as MacridVAE. There are concerns about the scalability of the approach to large-scale datasets and the complexity of real-world recommender systems with thousands of product facets.The paper is suited for acceptance as a poster, which allows for focused discussions and feedback that could help refine and further develop the approach.

**Justification For Why Not Higher Score:**

The paper is not accepted for an oral presentation due to its incremental nature and the limitations in addressing large-scale, real-world applications. The approach, while solid, does not represent a significant technical leap in the field, and concerns about its scalability and complexity in practical scenarios reduce its impact.

**Justification For Why Not Lower Score:**

This paper does make a valuable contribution to the field by advancing the understanding of multi-faceted user interest modeling. The methodological approach is solid, and the extensive experiments, along with the public availability of code and data, add to the paper’s merits. The incremental nature of the work does not diminish its value in contributing to the ongoing research in the field. Therefore, the paper merits acceptance in a poster format, which is an appropriate platform for presenting such work and receiving constructive feedback.

---

### Decision · Program_Chairs · 2024-01-16

Accept (poster)